# Poultry Litter Contamination by *Escherichia coli* Resistant to Critically Important Antimicrobials for Human and Animal Use and Risk for Public Health in Cameroon

**DOI:** 10.3390/antibiotics10040402

**Published:** 2021-04-08

**Authors:** Frédéric Moffo, Mohamed Moctar Mouliom Mouiche, Hervé Kapnang Djomgang, Patchely Tombe, Abel Wade, Fabrice Landjekpo Kochivi, Jarvis Bouna Dongmo, Cleophas Kahtita Mbah, Nabilah Pemi Mapiefou, Marie Paule Ngogang, Julius Awah-Ndukum

**Affiliations:** 1Department of Pharmacy, Pharmacology and Toxicology, School of Veterinary Medicine and Sciences, University of Ngaoundéré, Ngaoundere 454, Cameroon; mouichemoctar4@gmail.com (M.M.M.M.); kapnangherve@gmail.com (H.K.D.); patchtombe@gmail.com (P.T.); kochivifab@gmail.com (F.L.K.); dongmojarvis@gmail.com (J.B.D.); cleophaskaht@gmail.com (C.K.M.); peminabilah@yahoo.fr (N.P.M.); 2Laboratory of Animal Physiology and Health, Department of Animal Science, Faculty of Agronomy and Agricultural Sciences, University of Dschang, Dschang 479, Cameroon; awahndukum@yahoo.co.uk; 3National Veterinary Laboratory (LANAVET), Garoua 503, Cameroon; abelwade@gmail.com; 4Laboratoire de Recherche et d’Expertise Biomédicale (LABOREB), Yaoundé 35262, Cameroon; mngogang@gmail.com; 5Department of Animal Production Technology, College of Technology, University of Bamenda, Bambili 39, Cameroon

**Keywords:** *Escherichia coli*, poultry litter, antimicrobial resistance, prevalence, risk factors, public health, Cameroon

## Abstract

Residues of antimicrobials used in farm can exert selective pressure and accelerate the occurrence of multidrug resistant bacteria in litter. This study aimed to investigate the resistance profile of *Escherichia coli* isolated from poultry litter. A total of 101 *E. coli* strains was isolated from 229 litter samples collected and stored for two months in the laboratory at room temperature. Antimicrobial susceptibility testing was performed using the disk diffusion method. An overall resistance prevalence of 58.4% (95% CI: 48.8–68.0) was obtained with 59 *E. coli* strains resistant to various antimicrobial agents. High levels of resistance were observed with ciprofloxacin (21/59: 36%), imipenem (27/59: 45%), norfloxacin (44/59: 74%), ceftriaxone (44/59: 74%), and levofloxacin (44/59: 75%). These antimicrobials classified under the Watch group by WHO are indicators of the high AMR risk to public health in Cameroon. Multivariable logistic regression analysis revealed that a greater probability of high level of *E. coli* multidrug resistance was associated with lack of training in poultry farming (OR = 0.13, *p* = 0.01), less experience in poultry farming (OR = 11.66 *p* = 0.04), and the high frequency of digestive tract disease (OR = 0.10; *p* = 0.001). This study revealed that poultry litter constitutes a potential source of dissemination of resistant germs from farm animals to the environment and humans.

## 1. Introduction

Global livestock production has been growing rapidly and has moved increasingly towards industrialized systems where antimicrobial use (AMU) is an integral part of production [1,2]. The systematic use of antimicrobial may be due to the lack of appropriate biosecurity measures in farm which favors the emergence of pathogens [3]. However, the selective pressure exerted by the inappropriate use of antimicrobials has accelerated the emergence of antimicrobial resistance in both animal and human health especially in Low and Middle-Income Countries (LMICs) [4,5,6]. Emergence and spread of resistant bacteria from animals to humans through meat products and by-products and the environment have been reported [7,8]. Evident studies suggest that poultry environment especially litter is one of the major spreading pathways of multidrug resistant pathogens from animal to human [9]. In many LMICs, most poultry farms have no waste or litter treatment facilities, and it is often used as organic fertilizer or as feed supplements, especially in fish ponds [10,11]. This may increase the risk of exposure of antimicrobial resistant bacteria from the waste to human. Therefore, continued monitoring and surveillance of resistant pathogens across the human–animal and environmental interface is one of the best approaches for decision making and reduce AMR impact on public health [12]. According to the WHO’s Global Antimicrobial Resistance Surveillance System (GLASS), *Escherichia coli,* which is a ubiquitous bacterium, represents one of the best indicator for integrated AMR surveillance. It represents the most likely vehicles for the spread of resistance genes from animal to human and litter as a reservoirs of multidrug resistant *E. coli* from farm to the environment [13,14]. *E. coli* isolated from poultry litter in Nigeria [15], Ethiopia [16] Senegal [17], Belgium [9] and India [13] showed multidrug resistance prevalence ranging from 65% to 100%. In Cameroon, studies have been carried out on the assessment of drug resistance in bacteria isolated from cloacal samples, carcasses of both healthy and diseased animals and eggs [18,19,20,21], but little is known about the poultry environment and their contribution to the spreading of antimicrobial resistant bacteria to public health. Meanwhile reducing the load of resistant organisms released into the environment would significantly decrease the burden of resistant bacteria in all One Health settings, and thereby reduce the impact of AMR on public health. Therefore, the present study was carried out as part of the antimicrobial surveillance program to provide knowledge on the epidemiology of antimicrobial resistance and the factors contributing to the selection and spread of antimicrobial resistance germs in animal and environmental interface in Cameroon.

## 2. Results

### 2.1. Farmers Demographic Surveyed in Centre, Littoral and West Region of Cameroon

Out of the 229 farms investigated in the three regions of Cameroon, 118 (51.53%) were located in the West region, 70 (30.56%) in the Littoral region, and 41 (17.9%) in the Centre region. At least ¾ of the respondents were men and more than 85% of them had an age ranging between 30 and 60 years. Regarding education, nearly all of them (98%) have at least primary level of education but only 20% have received training in poultry farming. One-third (1/3) of the respondents have been in the activity for more than 5 years. Digestive tract (48.0%) and chronic respiratory tract (40.6%) infections were the most frequently reported compared to locomotory (9.6%) and nervous (1.8%) affections. In cases of unsuccessful treatment or persistence of disease after a treatment, ¾ of the respondents slaughtered and consumed their animals meanwhile a quarter of respondent took their birds to the market for sale. Half of them agreed that they consumed the dead carcasses meanwhile about 16% buried or incinerate the carcasses. As for litter management, more than half of the respondent saled the litter obtained from their systems while 40% spread it on their proper farms (Table 1).

The majority (75%) of the respondents purchased their antimicrobials from a veterinary pharmacy and were used for both preventive and curative purposes while 20% of respondents used antimicrobials as feed additives. Less than 10% of respondents returned the expired drugs to the pharmacy. Frequently used antimicrobials were oxytetracycline (50.2%), amoxicillin (13.1%), colistin (24.0%) and norfloxacin (6.1%) (Table 2).

### 2.2. Prevalence of Antimicrobial Resistance and Phenotypic Resistance Profile of Escherichia coli

Out of the 229 litter samples collected and stored for at least two months at laboratory temperature, a total of 101 samples tested positive to *E. coli* with a prevalence of isolation of 44.1% (95% CI: 37.7–50.5). After a susceptibility testing, 59 *E. coli* strains isolates were resistant to various antimicrobials with a global resistance prevalence of 58.4% (95% CI: 48.8–68.0). Moreover, 49/59 strains showed a resistance to at least three antibiotics for a multidrug resistance prevalence of 83.1% (95% CI: 73.5–92.6). Significant low levels of resistance were observed in the Littoral as compared to the West and Centre regions (*p* ˂ 0.05) (Table 3). Overall, levels of resistance observed with ciprofloxacin (21/59: 36%), imipenem (27/59: 45%) were lower than that observed with ampicillin (54/59: 91%), amoxicillin/clavulinic acid (53/59: 89%), and doxycycline (52/59: 88%) (Figure 1).

Regarding the WHO classification of critically important antimicrobials in human medicine, ampicillin, amoxicillin–clavulinic acid, and nalidixic acid were observed with resistance levels greater than 80% (47/59) (Figure 2). As for WHO Access—Watch—Reserve (AWaRe) categorization of antimicrobials, high levels of resistance were observed in antimicrobial in the Access group (45/59: 76%) follow by the Watch group (35/59: 59%) (Figure 3).

The multidrug resistance phenotypic patterns of all the *E. coli* isolates are shown in Table 4. The Multi-drug resistance index ranged from 0.07 to 1. The predominant MDR phenotype was AMC CEF AMP TET DOX NAL FLQ COT STR GEN NOR.

### 2.3. Risk Factors of Emergence and Diffusion of Resistant Germs from Poultry Litter

Univariable logistic regression analysis showed that region, training in poultry farming, frequency of digestive tract infections, experience in poultry farming and respect of vaccine protocol were associated (*p* < 0.25) with multidrug *E. coli* resistance rates. Multivariable logistic regression showed that the lack of training in poultry farming was significantly (OR = 0.13, *p* = 0.01) associated with high level of multidrug resistance. High frequency of digestive tract disease (OR = 0.10; *p* = 0.001), young farmers with at least five years of experience in poultry farming were significantly (OR = 11.66; *p* = 0.04) associated with high level of multidrug resistant *E. coli* in poultry litter (Table 5) in the study area.

## 3. Discussion

Based on the global public health implications of antimicrobial resistance, countries are increasingly being aware of the impact of AMR and are gradually taking measures to support the global fight. Monitoring and surveillance of antimicrobial resistance at the animal–human and environmental interface may help to reduce the transfer of AMR from animals to humans directly or indirectly through the environment [22]. In this interface, manure from food-producing animal account for most important reservoirs of maintenance and spread of resistant bacteria and resistant gene’s element [13]. Litter which represents a mixture of poultry feces, feed, contaminated water, and different bedding materials accumulated during farming may become a potential source of infectious agents and *E. coli* is used as an indicator bacteria to detect their possible presence [23]. *E. coli* as a ubiquitous commensal germ, present in human and animal intestinal flora, can easily be excreted into the environment. During farming, antimicrobials are used for curative or preventive purposes and also used as growth promoters on poultry farms, thus the spread of resistant of *E. coli* [24]. Heuer and Smalla [25] reported that the use of large amounts of antimicrobials in poultry farming may result in an increased amount of resistant bacteria in poultry, their excreta and consequently in the litter and environment.

In Cameroon like other LMICs, the bedding material obtained is generally spread on farms or sold to crop farmers without a being treated. The present study was initiated with the assumption that the storage of litter in plastic bags at room temperature for at least two months may help to reduce potential pathogens and this period approximately correspond to the delay period observed by most farmers before application of litter as, manure. It appears that isolation frequency of 45.41% was lower than 59.1% reported in the fresh broiler litter in previous studies in Cameroon [26] and 58.0% in Zambia [27]. *E. coli* is a commensal bacteria present in the intestinal tract of chicken and poultry environment, and hence storage of samples may have induced the destruction of some fragile strains. A susceptibility testing of *E. coli* strains gave an overall prevalence of resistance (58%) comparable to the 50.3% reported by Mouiche et al. [22] in a systematic review and meta-analysis in Cameroon, but lower than 94% reported by Cookey et al. [15] in Nigeria and 100% Eyasu et al. [16] in Ethiopia, respectively. At least 49 (83.1%) of the isolated strains were multidrug resistant and exhibited resistance to at-least three or more antimicrobials. The resistance rate observed with ciprofloxacin (36%) was similar to 42% reported by Adelowo et al. [28] in Nigeria, lower than 57% reported by Louokdom et al. [29] in Cameroon, but higher than 13% reported by Vounba et al. [17] in Senegal and 22% by Nfongeh et al. [30] in Nigeria. As for imipenem, the resistance rate observed in this study was higher than 4% reported by Phiri et al. [27] in Zambia. The emergence of carbapenem resistance is alarming as the World Health Organization classifies these molecules as critically important antimicrobials and carbapenems are the last-resort antimicrobials for treating a wide range of infections caused by multidrug-resistant Gram-negative bacteria [27]. Resistance rate of *E. coli* to ceftriaxone (74%) in this study was higher than 18.9% reported by Abdalla et al. [31] in South Africa and 59% reported by Ngogang et al. [26] in Cameroon while, 75% of resistance observed with levofloxacin was higher than 45.63% reported by Tchapa and Chapagain [32] in Nepal. The observed levels of resistance to imipenem, ceftriaxone and levofloxacin were however unexpected since these antimicrobials are not used in animal production in Cameroon. Thus, these levels of resistance might suggest that factors other than antimicrobial use may be contributing to the selection of resistance among the present isolates. The higher resistance rate of *E. coli* to these antimicrobials of critically importance for human medicine may be an indication of the development of co-resistance. Inappropriate use of antimicrobial in farm represents a selective pressure for resistant bacteria which can develop co-resistance and cross-resistance between several classes of antimicrobials [33]. In addition, norfloxacin and ciprofloxacin were reported to be mostly used in poultry production in Cameroon [34,35] and might act as selective pressure for the development of co-resistance and cross-resistance to others antimicrobials of the same class [33]. Highly resistance rate of *E. coli* observed with the critically antimicrobial agents in human indicates a high level of exposure of public health to AMR since imipenem, ceftriaxone and levofloxacin are used as second option line treatment in hospital settings in Cameroon [22,26]. Respect of the regulation code on the use of antimicrobials in animal production in Cameroon [36], continued sensitization of farmers towards the consequences of widespread use of substandard drugs [37] coupled with the improvement of farmer’s knowledge towards good usage of antimicrobial and the respect of biosecurity measures in farm [35] might help to mitigate the impact of AMR on public health in Cameroon.

The multidrug resistance of *E. coli* to fourteen antimicrobials tested was in line with the emergence of a global threat concerning the development of high levels of antimicrobial resistance to multiple classes of drugs [38,39]. MDR index trends up to one from certain strains implies isolates from high-risk contaminated sources with frequently antimicrobial use. Similar reports by Chen and Jiang [40] in US indicates multidrug resistant *E. coli* isolated from broiler litter to at least seven antimicrobials. This may be correlated with the indiscriminate use of antimicrobials in poultry farm for prophylaxis purposes as observed in this study and their excretion in the litter. Kumar et al. [41] reported that chicken excreted 75–80% of tetracyclines, 60% lincosamides, and 50% to 90% macrolides in the feces. Fluoroquinolones and sulfonamides were also recorded to be highly excreted in the litter [42] and this may explain the high level of antimicrobial resistance observed within or between antimicrobial classes. Moreover, auto-prescription of antimicrobials from open market may also escalate the emergence of AMR. Drugs from parallel markets are often of substandard and poor-quality. Evidence suggest that poor-quality medicines provide subtherapeutic doses of active pharmaceutical ingredients, resulting from inadequate amounts of pharmaceutical, ineffective release, presence of impurities or degradation of compounds, are believed to contribute to antimicrobial resistance by exposing microbes to a level of antimicrobial that will not effectively kill the whole microbial population [4,6]. Such practice of using antimicrobials by untrained farmers for treatment of chickens without proper diagnosis and strict adherence to proper dosage and frequency of administration, could result in AMR development and thereby increase the risk to public health. Importunately, results in this study revealed an increasing resistance to all antimicrobial classes, including critically important antimicrobials for human use and the watch and reserve categories of antimicrobials leading to serious concern to human health [23]. Hence, this reiterates the call for a holistic review on the use of antimicrobials as growth promoters in the food–animal production chain.

Factors associated with multidrug resistant *E*. *coli* in this study includes the lack of training in poultry farming, the high frequency of digestive tract disease in farm and fewer experience in poultry farming. These observations were in line with the fact that the knowledge and farmer’s behaviour can significantly influence their decision to use antimicrobials and thus emergence of AMR in farm [43,44]. The high level of resistance obtained in this study is an indication that litter can serve as a reservoir for resistant genes and AMR organisms capable of being transmitted to humans even long after their extraction from farms. Hence, more precautions must be taken to preserve public health from related risks associated with the reuse of litter from animal farms, especially when the bedding material in poultry farms are removed and either spread on farms or sold to crop farmers without previous treatment.

## 4. Materials and Methods

### 4.1. Study Site and Study Design

A cross-sectional survey was carried out from June to November 2019 in the Centre (3°16′0″–6°26′0″ N 10°40′0″–13°21′0″ E), Littoral (4°03′–4°90′ LN and 9°42′–10°43′ LE) and West (5°25′0″–5°35′0″ N 10°20′0″–10°35′0″ E) regions of Cameroon (Figure 4). Centre (14.3%), Littoral (10%) and West (36.8%) regions contribute over 61.1% of broilers and layers production in the country [45]. These poultry producing regions can constitute good AMR sentinel surveillance sites for the country. A minimum sample size of 186 was estimated [46] based on previous reports on the prevalence of *E. coli* in litter of 86% [17] with a confidence interval of 95% and precision set at 5%. Broiler farms with a minimum of 500 chickens per selected farm were included in the study. The study used a stratified random technique. Sampling of each region was according to their proportion in the national flock size (West, 53%, Centre, 35%, and 12% in the littoral regions). Random number generation technique was used for the selection of farms from a list of poultry farmers obtained at the Delegations of Livestock, Fisheries and Animal Industries (DREPIA) in the study regions and completed by private field veterinary practitioners. The scientific research and ethics committee of the School of Veterinary Medicine and Sciences of the University of Ngaoundere-Cameroon (2019/017/UN/ESMV/D) provided ethical approval for this research. The regional delegations in charge of animal health permitted the survey in the four regions [Centre (000083/L/MINEPIA/SG/DREPIA-CE), Littoral (079/L/RDREPIA-LT) and West (N°25/19/LDREPIA-O/SRAG)]. Following explanation of the purpose of study to poultry farmers in these regions, farmers provided written consents before they and their farms were included in the survey. A semi-structured questionnaire was used to collect data on socio-demographic characteristics of farmers and farm characteristics in the three regions. Demographic characteristics of poultry farmers included gender, age, education, training, and experience in poultry farming. Farm characteristics included farm size, stocking density (number of chickens per m^2^) and the management of poultry litter. Ten farmers randomly selected within the study regions were used to test the clarity, reliability, and validity of the questionnaire.

### 4.2. Sample Collection, Processing and Escherichia coli Isolation

In each farm, 40–50 g of poultry litter was collected in sterile plastic bags. The collection was done in 3 spots in the breeding hall (also in each breeding hall, when the farm had several halls) and mixed for homogeneity. The sample were carried at the National Veterinary laboratory (LANAVET) Annex Yaoundé and stored in laboratory at room temperature in plastics bags for 2 months before further processing. This period approximately correspond to the delay period observed by most farmers before application of litter as, manure. In the laboratory, the samples were individually diluted in 10 mL of peptone water for enrichment, crushed, vortexed and filtered to remove debris and other solid materials. A minimum of 5.0 mL of the suspension from filtered medium was introduced in cryotubes and stored at +4 °C for *E. coli* isolation and identification. An aliquot of 1 mL of each fecal suspension was mixed with glycerol (15% final concentration) and stored at −80 °C for long-term preservation of the original samples. The fecal suspensions were further diluted (1:10) with sterile distilled water and used for isolation of bacteria. Culture and isolation of *Escherichia coli* was performed using standard media according to the manufacturers’ instructions and essentially as previously described ISO [47]. A sterile glass rod was used to spread 30 µL of diluted suspension onto 100 mm diameter MacConkey agar plates (Biolife Italiana S.r.l., Milan, Italy) which were then incubated for 24 h at 37 °C. The next day pink colored presumptive *E. coli* colonies were sub-cultured onto nutrient agar (Biolife Italiana S.r.l., Milan, Italy) and incubated for 24 h at 37 °C. Five presumptive *E. coli* colonies were randomly selected and transferred onto nutrient agar for further identification using biochemical tests (hydrogen sulfide production, carbohydrate fermentation, urease test, methyl red test, motility test, and indole test). Colonies fulfilling the preceding criteria were further characterized using API^®^ 20 E gallery (bioMérieux, Lyon, France).

### 4.3. Antimicrobial Susceptibility Test

The antimicrobial susceptibility testing of *E. coli* isolates was conducted using disk diffusion method and interpreted according to breakpoints as defined by the European Committee on Antimicrobial Susceptibility Testing (EUCAST) [48]. The following antimicrobials from BD Sensi-Disc^TM^ (Heidelberg, Germany) (Table 6) was chosen based on the farm investigation and the importance of such drugs for human health according to WHO criteria [49] and the WHO AWaRe (Access, Watch, and Reserve) Categorization [50]. Strains with an intermediate susceptible result were considered resistant. A multidrug resistant (MDR) strain was defined as a strain resistant to at least three different antimicrobial agents [7]. A farm was defined as “positive” for a resistant *E. coli* if at least one isolate resistant to the antimicrobial drug under study was isolated from the farm. Quality controls for identification and susceptibility testing were performed on a weekly basis according to EUCAST guidelines.

### 4.4. Data Analysis

Data entry was performed with Microsoft Excel 2013 (Microsoft Corporation, Redmond, WA, USA). Descriptive statistics comprising percentages were used to indicate the proportion of poultry farms investigated and prevalence of resistance to antimicrobials amongst *E. coli* isolates in these three regions. As for the antimicrobial use, multiple antimicrobial agent could be selected by the farmers. The frequency of use was calculated as the ratio of the number of times a substance was selected by the total number of times all the substance was reported. Multi-drug resistance (MDR) was recorded if one isolated strain was resistant to at least three antimicrobial agents. MDR index was measured as the total number of antimicrobials to which the test isolates depicted resistance over the total number of antimicrobials to which the test isolate has been evaluated for susceptibility. The association of potential risk factors with *E. coli* multidrug resistance prevalence was analyzed using multiple logistic regression. Stratification method was used for those variables showing significant association to see any difference between the crude and adjusted results. Then, after further checking for collinearity, variables with *p*-value less than 0.25 during univariable analysis were further analyzed using a multivariable logistic regression model. Odds ratio was used to see degree of association and confidence level was held at 95% and significance was at *p* < 0.05. All data were computed using IMB SPSS Statistics (ver. 20.0).

## 5. Conclusions

The present study highlights persistence of *E. coli* in poultry litter even weeks after removal from the breeding hall. High multidrug resistance prevalence to critically antimicrobial agents for human medicine was observed. Improving biosecurity measures in farms is necessary to avoid the entry of pathogens or their dissemination within the farms and environment. Sensitization and education through campaigns, trainings and other accessible media communication tools will empower farmers and enhance their knowledge on antimicrobial use, thus contributing to reduce the burden of AMR in public health in Cameroon.

## Figures and Tables

**Figure 1 antibiotics-10-00402-f001:**
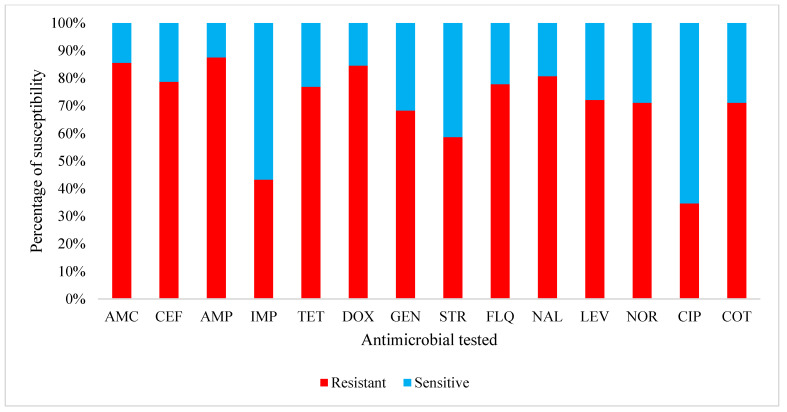
Sensitivity of multidrug resistance *Escherichia coli* strains isolated (*n* = 59) in poultry litter to common antimicrobials used in the Centre, Littoral, and West Regions of Cameroon. AMC: amoxicillin–clavulanic acid; AMP: ampicillin; CEF: ceftriaxone; CIP: ciprofloxacin; COT: cotrimoxazole; DOX: doxycycline; FLQ: flumequine; GEN: gentamycin; IMP: Imipenem; LEV: levofloxacin; NAL: nalidixic acid; NOR: norfloxacin; STR: streptomycin; TET: tetracycline.

**Figure 2 antibiotics-10-00402-f002:**
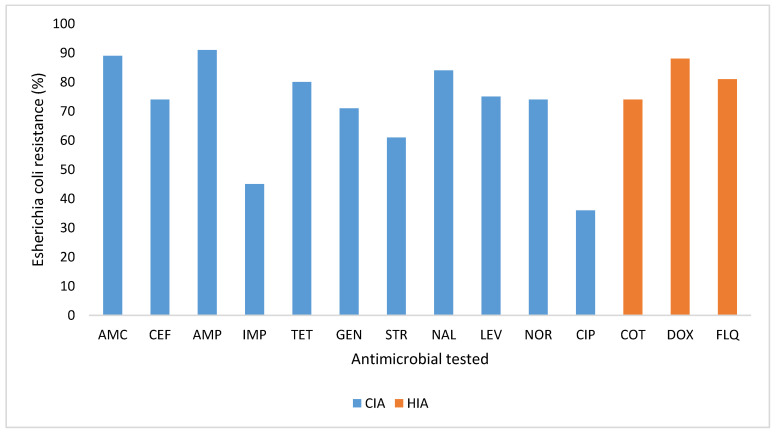
Multidrug resistance profile of *Escherichia coli* isolated (*n* = 59) in poultry litter to common antimicrobials with respect to the critically antimicrobial agents in human medicine in the Centre, Littoral, and West Regions of Cameroon. CIA: Critically important antimicrobial agents in human Medicine; HIA: Highly Important Antimicrobial agents in human medicine; IA: Important Antimicrobials agents in human medicine; AMP: Ampicillin; AMC: amoxicillin–clavulanic acid; CEF: ceftriaxone; CIP: Ciprofloxacin; TET: Tetracycline; GEN: Gentamicin; NOR: Norfloxacin; IMP: Imipenem; LEV: levofloxacin; STR: Streptomycin; NA: Nalidixic acid; FLQ: flumequine; COT: Cotrimoxazole; Dox: Doxycycline.

**Figure 3 antibiotics-10-00402-f003:**
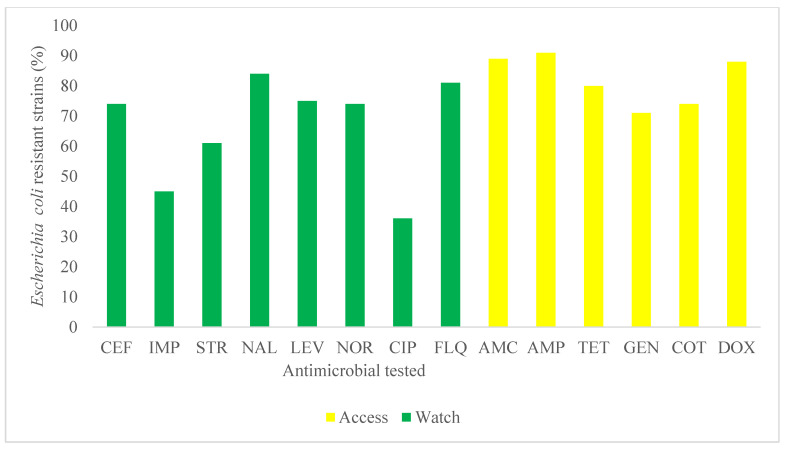
Multidrug resistance profile of *Escherichia coli* isolated (*n* = 59) in poultry litter to common antimicrobials with respect to the WHO Access—Watch—Reserve (AWaRe) categorization of antimicrobials in human medicine in the Centre, Littoral and West Regions of Cameroon. AMP: Ampicillin; AMC: amoxicillin–clavulanic acid; CEF: ceftriaxone; CIP: Ciprofloxacin; TET: Tetracycline; GEN: Gentamicin; NOR: Norfloxacin; IMP: Imipenem; LEV: levofloxacin; STR: Streptomycin; NA: Nalidixic acid; FLQ: flumequine; COT: Cotrimoxazole; Dox: Doxycycline.

**Figure 4 antibiotics-10-00402-f004:**
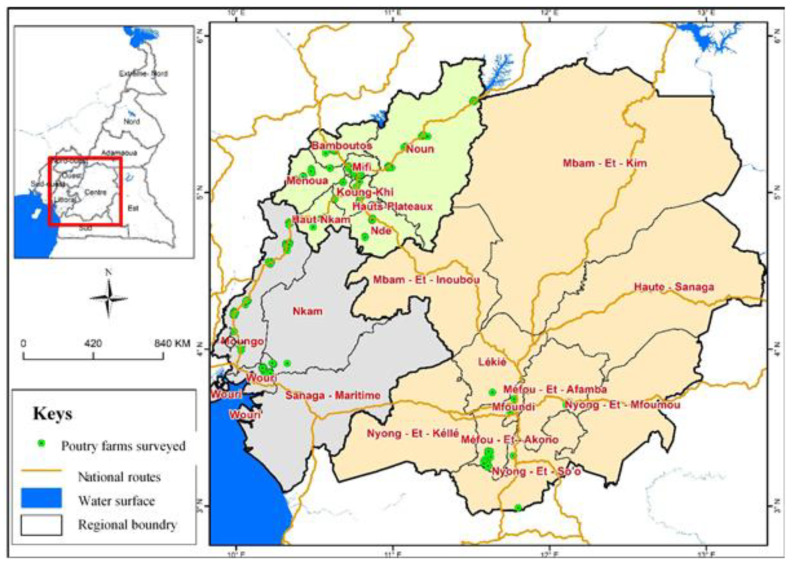
Geographical distribution of the poultry farms surveys in the Centre, Littoral, and West region of Cameroon.

**Table 1 antibiotics-10-00402-t001:** Demographic and poultry farm characteristics surveyed (*n* = 229) in the West, Littoral and Centre regions of Cameroon.

	Regions			
Socio Professional Characteristics	West	Littoral	Centre	Total
	*n* = 118 (%)	*n* = 70 (%)	*n* = 41 (%)	*n* = 229 (%)
Gender				
Male	90 (76.3)	44 (62.9)	26 (63.4)	160 (69.9)
Female	28 (23.7)	26 (37.1)	15 (36.6)	69 (30.1)
Age range (Year)				
20–29	3 (2.5)	5 (7.1)	7 (17.1)	15 (6.6)
30–39	35 (29.7)	23 (32.9)	12 (17.1)	70 (30.6)
40–49	65 (55.1)	29 (41.4)	9 (22.0)	103 (45.0)
50–59	15 (12.7)	13 (18.5)	13 (31.7)	41 (17.9)
Educational level				
None	2 (2.9)	2 (2.9)	0 (0.0)	4 (1.7)
Primary	12 (10.2)	23 (32.9)	3 (7.3)	38 (16.6)
Secondary	70 (59.3)	29 (41.4)	25 (61.0)	124 (54.1)
Higher	34 (28.8)	15 (21.4)	11 (26.8)	60 (26.2)
Training in poultry farming				
Yes	33 (27.7)	9 (12.9)	10 (24.4)	52 (22.7)
No	85 (72.0)	61 (87.0)	31 (75.6)	177 (77.0)
Duration in poultry farming (month)				
0–5	54 (45.8)	22 (31.4)	28 (68.3)	104 (68.3)
6–11	58 (49.2)	28 (40.0)	11 (26.8)	97 (26.8)
12–17	4 (3.4)	16 (22.9)	1 (2.4)	21 (2.4)
≥18	2 (1.7)	4 (5.7)	1 (2.4)	7 (2.4)
Training on antimicrobial use				
Yes	30 (25.4)	2 (4.9)	4 (5.7)	36 (15.7)
No	88 (74.6)	39 (95.1)	66 (94.3)	193 (84.3)
Respect of vaccinal protocols				
No	16 (13.6)	1 (1.4)	0 (0.0)	17 (7.4)
Partially	34 (28.8)	1 (1.4)	1 (2.4)	36 (15.7)
Totally	68 (57.6)	68 (97.1)	40 (97.0)	176 (76.9)
Flock size				
<1000	43 (36.4)	24 (34.3)	17 (41.5)	84 (36.7)
1001–5000	68 (57.6)	36 (51.4)	18 (43.9)	122 (53.3)
5001–10,000	7 (5.9)	10 (14.3)	3 (14.6)	23 (10.0)
Stocking density (number of chicken/m^2^)				
<6	24 (20.3)	34 (48.6)	2 (4.9)	60 (26.2)
9–6	46 (38.9)	4 (5.7)	25(60.9)	75 (32.8)
>9	48 (40.7)	32 (45.7)	14 (34.2)	94 (41.1)
Sanitary void lap (days)				
≤14	11 (9.3)	5 (7.1)	2 (4.9)	18 (7.9)
14	16 (13.6)	18 (25.7)	3 (7.3)	37 (16.2)
14–30	42 (35.6)	39 (55.7)	36 (87.8)	117 (51.1)
≥30	49 (41.5)	8 (11.4)	0 (0.0)	57 (24.9)
Frequency of disease				
Digestive	36 (32.6)	49 (44.6)	26 (22.9)	110 (48.0)
Respiratory	19 (20.3)	41 (44.3)	33 (36.6)	93 (40.6)
Nervous	3 (2.2)	1 (1.4)	0 (0.0)	4 (1.8)
Locomotory	18 (15.5)	0 (0.0)	4 (9.7)	22 (9.6)
Case of unsuccessful treatment				
Sale	24 (20.3)	13 (18.0)	16 (39.0)	53 (23.1)
Slaughter and consume	93 (78.8)	56 (80.0)	25 (61)	174 (76)
Quarantine	1 (0.8)	1 (1.4)	2 (0.9)	2 (0.9)
Management of death chicken				
Consummed	37 (31.4)	58 (82.9)	18 (43.9)	111 (49.3)
Dispose in the dustbin	56 (47.5)	8 (11.9)	15 (36.6)	79 (34.5)
Incinerate or bury	25 (21.2)	4 (5.7)	8 (19.5)	37 (16.2)
Litter management				
Sale to crops farmers	49 (41.5)	52 (74.3)	22 (53.7)	123 (53.7)
Spread on farms	69 (58.5)	18 (25.7)	19 (46.3)	106 (46.3)

**Table 2 antibiotics-10-00402-t002:** Management of antimicrobial use in poultry farm surveyed (*n* = 229) in the West, Littoral, and Centre regions of Cameroon.

	Regions			
Factors	West	Littoral	Centre	Total
Purchase of antimicrobial				
Veterinary pharmacy	86 (72.9%)	50 (71.4%)	39 (95.1%)	175 (76.4%)
Parallel market	32 (27.1%)	20 (28.0%)	2 (4.9%)	54 (23.6%)
Reasons for AMU				
Preventive	13 (11.0%)	7 (10%)	1 (2.4%)	21 (9.2%)
Curative	13 (11.0%)	1 (2.4%)	11 (15.7%)	25 (10.9%)
Preventive and curative	92 (77.9%)	52 (74.3%)	39 (95.1%)	183 (79.9%)
Addition of antimicrobial in feed				
Yes	28 (23.7%)	16 (22.9%)	2 (4.9%)	46 (20.1%)
No	90 (76.3%)	54 (77.1%)	39 (95.1%)	183 (79.9%)
Management of expired drug				
Throw away in environment	15 (56.8%)	30 (42.9%)	67 (36.6%)	112 (48.9%)
Return to pharmacy	14 (11.9%)	8 (11.4%)	0 (0.0%)	22 (9.6%)
Administrated to animal	26 (31.4%)	32 (45.7%)	37 (63.4%)	95 (41.5%)
Most common AMU used in surveyed Farms				
Amoxicillin	22 (18.6%)	5 (7.1%)	3 (7.3%)	30 (13.1%)
Colistin	18 (15.7%)	11 (15.7%)	26 (63.4%)	55 (24.0%)
Doxycycline	3 (2.5%)	2 (2.9%)	4 (9.8%)	9 (3.9%)
Enrofloxacin	0 (0.0%)	2 (2.9%)	1 (2.4%)	3 (1.3%)
Flumequine	0 (0.0%)	2 (2.9%)	1 (2.4%)	3 (1.3%)
Norfloxacin	0 (0.0%)	14 (20.0%)	0 (0.0%)	14 (6.1%)
Oxytetracycline	75 (50.2%)	34 (48.6%)	6 (14.6%)	115 (50.2%)

AMU: antimicrobial use.

**Table 3 antibiotics-10-00402-t003:** Prevalence of resistance of *E. coli* isolated in the Centre, Littoral, and West regions of Cameroon.

Regions	Number of Sample Collected	Number of *E. coli* Isolated (%)	*p*-Value	Number of Strains Resistant and Prevalence of Resistance (%) 95% CI	*p*-Value
West Littoral Centre	118 70 41	61 (51.7%) 20 (28.6%) 20 (48.8%)	0.66	40 (65.6% (53.6–77.5)) 6 (30.0% (9.9–50.1)) 13 (65.0% (44.0–85.9))	0.0062
Total	229	101 (44.1%)		59 (58.4% (48.8–68.0))	

**Table 4 antibiotics-10-00402-t004:** Phenotypic resistance profile of *E. coli* isolated (*n* = 59) in poultry litter in Centre, Littoral, and west region of Cameroon and to tested antimicrobials.

Number of Antibiotic	Phenotypic Resistance Profile	Number of Isolates	MDRI
1	CEF	5	0.07
	AMP	3	0.07
2	CEF NAL	2	0.14
5	AMC CEF AMP TET DOX	5	0.36
6	AMC CEF AMP TET DOX NAL	5	0.43
7	AMC CEF AMP TET DOX NAL FLQ	3	0.50
	AMC CEF AMP STR NOR NAL FLQ	2	0.50
8	AMC CEF AMP DOX NAL NOR COT LEV	5	0.57
	AMC CEF AMP DOX NAL NOR TET STR	2	0.57
9	AMC CEF AMP TET DOX NAL FLQ COT STR	7	0.64
10	AMC CEF AMP TET DOX NAL FLQ COT STR LEV	4	0.71
11	AMC CEF AMP TET DOX NAL FLQ COT STR GEN NOR	12	0.79
12	AMC CEF AMP TET DOX NAL FLQ COT STR GEN NOR LEV	2	0.86
13	AMC CEF AMP TET DOX NAL FLQ COT STR GEN NOR LEV IMP	1	0.93
14	AMC CEF AMP TET DOX NAL FLQ COT STR GEN NOR LEV IMP CIP	1	1

MDRI = Multidrug resistance index; AMP: Ampicillin; AMC: amoxicillin–clavulanic acid; CEF: ceftriaxone; CIP: Ciprofloxacin; TET: Tetracycline; GEN: Gentamicin; NOR: Norfloxacin; IMP: Imipenem; STR: Streptomycin; NA: Nalidixic acid; FLQ: flumequine; COT: Cotrimoxazole; Dox: Doxycycline.

**Table 5 antibiotics-10-00402-t005:** Logistic regression analyses of the risk factors for multidrug resistant *E. coli* isolated in poultry litter in Centre, Littoral, and West region of Cameroon.

Risk Factor	Category	Number of Samples Tested	Univariable		Multivariable	
			OR (95% CI)	*p* Value	OR (95% CI)	*p* Value
Region	West	61 (60.4)	2.19 (0.68–7.01)	0.2	0.98 (0.21–4.69)	0.98
	Littoral	20 (19.8)	0.23 (0.06–0.87)	0.03	0.25 (0.06–1.13)	0.07
	Centre	20 (19.8)	1.0		1.0	
Training in poultry farming	No	76 (75.2)	0.39 (0.12–1.25)	0.11	0.13 (0.03–0.64)	0.01
	Yes	25 (24.8)	1.0		1.0	
Frequency of digestive tract infections	High	55 (54.5)	0.11 (0.04–0.40)	0.001	0.10 (0.02–0.41)	0.001
	Low	46 (45.5)	1.0		1.0	
Experience in poultry farming (months)	0–5	45 (44.6)	4.67 (0.89–27.40)	0.06	11.66 (1.12–121.10)	0.04
	6–11	49 (48.5)	3.02 (0.60–15.20)	0.18	6.37 (0.66–61.66)	0.11
	12–17	7 (6.9)	1.0		1.0	
Respect of vaccinal protocol	No	10 (9.9)	5.52 (0.66–46.05)	0.11	1.14 (0.10–13.67)	0.92
	Partially	20 (19.8)	11.66 (1.48–92.14)	0.02	5.11 (0.52–49.95)	0.16
	Totally	71 (70.3)	1.0		1.0	

**Table 6 antibiotics-10-00402-t006:** List of antimicrobials used for susceptibility testing.

WHO Classification	Antimicrobial Agents	Class	WHO-AWaRe	Disc Charges
Critically Important Antimicrobials	Ampicillin	Penicillin	Access	10 µg
	Amoxicilin/clavualanic acid	Penicillin		20/10 µg
	Gentamicin	Aminoglycoside		10 µg
	Ceftriaxone	Cephalosporine	Watch	30 µg
	Ciprofloxacin	fluoroquinolone		30 µg
	Norfloxacin	fluoroquinolone		10 µg
	Streptomycin	Aminoglycoside		10 µg
	Flumequine	Quinolone		30 µg
	Imipenem	Carbamate		10 µg
	Levofloxacin	fluroquinolones		5 µg
	Nalidixic acid	fluoroquinolone		30 µg
Highly Important antimicrobials	Tetracycline	Tetracycline	Access	30 µg
	Doxycycline	Tetracycline		30 µg
	Cotrimoxazole	Diaminopyrimidine/sulphamide		25 µg

## Data Availability

The datasets used and/or analyzed during the current study are available from the corresponding author upon reasonable request.

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
