# Peer review of "Poultry Litter Contamination by Escherichia coli Resistant to Critically Important Antimicrobials for Human and Animal Use and Risk for Public Health in Cameroon"

_antibiotics, 2021, doi:10.3390/antibiotics10040402_

Round 1
Reviewer 1 Report
The manuscript by Moffo et al. describes the assessment of the prevalence of antibiotic-resistant E. coli isolates in poultry farm litter in Cameroon, and its association with various farm-related factors. The spread of the antibiotic-resistant bacterial strains is an important problem. It is crucial to monitor the prevalence of resistant strains, both to understand its sources (e.g. poultry farms), and to see if the introduced measures are having any effect. Thus, the presented manuscript provides important information and, I believe, will be interesting to the readers and useful for subsequent meta-analyses and reviews.
In general, the manuscript is well written. However, in my opinion, the following points need to be clarified:
- It is not immediately clear what exactly is meant by “overall resistance prevalence” (58.4%) and “multidrug resistance prevalence” (69.3%). It is confusing that the latter is greater than the former. I think it should be explained in the text where it is first mentioned (page 5).
- When percentages are given, it is not always clear what they mean. For example, Table 4 states that there were 101 isolates in total, of which 59 were resistant (58.4%, the overall resistance prevalence). However, the text on the same page says there were 91% strains resistant to ampicillin. Are they 91% of all strains? Of resistant strains? This should be explicitly stated to avoid confusion. The same is true for the abstract and Figures 2, 3, and 4.
- Table 2 (which is in fact the first table in the text – I guess it is because the Methods section was placed before the Results in some previous version of the manuscript) is not sufficiently explained. What do the numbers in parentheses mean? It appears that they are percentages, however, % sign in only present for some numbers, but not the others. The table caption should explain what the numbers mean.
- On page 7 the text says “All the isolates were resistant to at least two antimicrobial agents”. However, only 59 of 101 isolates were resistant. Should it be “All the resistant isolates were resistant to at least two antimicrobial agents”?
Author Response
In general, the manuscript is well written. However, in my opinion, the following points need to be clarified:
- It is not immediately clear what exactly is meant by “overall resistance prevalence” (58.4%) and “multidrug resistance prevalence” (69.3%). It is confusing that the latter is greater than the former. I think it should be explained in the text where it is first mentioned (page 5).
Re: After a susceptibility testing, 59 E. coli strains isolates were resistant to various antimicrobials. This gave a global resistance prevalence of 58.4% [95% CI: 48.8-68.0]. However, a multidrug resistance prevalence of 83.1% [95% CI :73.5-92.6] was observed. The multi-drug prevalence represented the number of resistant strains to at least three antibiotic over the total number of resistant strains (49/59).
- When percentages are given, it is not always clear what they mean. For example, Table 4 states that there were 101 isolates in total, of which 59 were resistant (58.4%, the overall resistance prevalence). However, the text on the same page says there were 91% strains resistant to ampicillin. Are they 91% of all strains? Of resistant strains? This should be explicitly stated to avoid confusion. The same is true for the abstract and Figures 2, 3, and 4.
Re: This percentage represented the number of resistant strain per antimicrobial agent or group of antimicrobial divided the total number of resistant strain. In the text, this was modified by adding the number of strains per individual molecule or group of molecules.
- Table 2 (which is in fact the first table in the text – I guess it is because the Methods section was placed before the Results in some previous version of the manuscript) is not sufficiently explained. What do the numbers in parentheses mean? It appears that they are percentages, however, % sign in only present for some numbers, but not the others. The table caption should explain what the numbers mean.
- Re: Presentation of the number in parentheses was an error. This was modified and the % sign was deleted within table 2
- On page 7 the text says “All the isolates were resistant to at least two antimicrobial agents”. However, only 59 of 101 isolates were resistant. Should it be “All the resistant isolates were resistant to at least two antimicrobial agents”?
Re: This is referring to the resistant strains not all of the 101 strains isolated in the study
Reviewer 2 Report
This is a descriptive study to determine the prevalence of resistant E. Coli in poultry litters in three geographical areas of Cameroon. The report also includes antibiotic resistance profile with most E. Coli being resistant to two antibiotics. Logistic regression analysis of survey data revealed that E. Coli multidrug resistant was associated with lack of training and experience in poultry farming
and the high frequency of digestive tract diseases. The authors concluded that poultry
litter constitutes a potential source of dissemination of resistant germs from farm animals
to the environment and humans.
While the study findings are not novel in the sense that similar observations have been made in other comparable countries like Cameroon, the study nevertheless adds more evidence to the growing recognition that poultry litters is a source of AMR. In addition, it is imperative that poultry farmers need to know appropriate and judicious use of antibiotics. The scientific reason for storing the sample at room temperature for two months is missing.
The following comments are to help the authors improve the manuscript.
- The manuscript needs editorial help.
- Unless required by the journal, please number the tables and figures in the order they appear in the manuscript. This makes it easier to follow the manuscript.
- Table 2 is rather large. Is it possible to divide it up into more relevant components?
- Table 2. Please explain what is meant by “Respect of vaccinal protocol” and the implication of the finding.
- Page 4 text above “Table 3”: As for litter management, more than the majority sell it to agricultural farmers while 40% spread it on their proper farms (Table 3). Should this refer to table 2? Data presented in Table 3 has not been mentioned in the text.
- Levofloxacin (LEV) is missing in the legend.
- Page 10. The statement, “The present study was initiated with assumption that the storage of litter in plastic bags at room temperature for at least two months may help to reduce potential pathogens.” Needs further elaboration. If this storage at room temperature was intentional, then more rationale is needed. Also, what was the room temperature during the storage? If the purpose is to find out the effect of storage at room temperature, the study should have included samples stored at -80oC for comparison.
Author Response
Comments and Suggestions for Authors
This is a descriptive study to determine the prevalence of resistant E. Coli in poultry litters in three geographical areas of Cameroon. The report also includes antibiotic resistance profile with most E. Coli being resistant to two antibiotics. Logistic regression analysis of survey data revealed that E. Coli multidrug resistant was associated with lack of training and experience in poultry farming
and the high frequency of digestive tract diseases. The authors concluded that poultry
litter constitutes a potential source of dissemination of resistant germs from farm animals to the environment and humans.
While the study findings are not novel in the sense that similar observations have been made in other comparable countries like Cameroon, the study nevertheless adds more evidence to the growing recognition that poultry litters is a source of AMR. In addition, it is imperative that poultry farmers need to know appropriate and judicious use of antibiotics. The scientific reason for storing the sample at room temperature for two months is missing.
The following comments are to help the authors improve the manuscript.
1. The manuscript needs editorial help.
2. Unless required by the journal, please number the tables and figures in the order they appear in the manuscript. This makes it easier to follow the manuscript.
Tables and figures were numbered chronologically as they appears in the text
3. Table 2 is rather large. Is it possible to divide it up into more relevant components?
4. Table 2. Please explain what is meant by “Respect of vaccinal protocol” and the implication of the finding.
During farming, vaccines are used to prevent most of the viral infectious diseases in poultry production. Therefore, the respect of vaccinal protocol as indicated by the national program on prevention and eradication of disease might help reduce the emergence of pathogens and use of antimicrobial agent to treat opportunistic diseases.
5. Page 4 text above “Table 3”: As for litter management, more than the majority sell it to agricultural farmers while 40% spread it on their proper farms (Table 3). Should this refer to table 2? Data presented in Table 3 has not been mentioned in the text.
The majority (75%) of the respondents purchased their antimicrobials from a veterinary pharmacy and were used for both preventive and curative purposes while 20% of respondents used antimicrobials as feed additives. Less than 10% of respondents reported returned the expired drugs to the pharmacy. Frequently used antimicrobials were oxytetracycline (50.2%), amoxicillin (13.1%), colistin (24.0%) and norfloxacin (6.1%) (Table 3).
6. Levofloxacin (LEV) is missing in the legend.
This was added in the manuscript
7. Page 10. The statement, “The present study was initiated with assumption that the storage of litter in plastic bags at room temperature for at least two months may help to reduce potential pathogens.” Needs further elaboration. If this storage at room temperature was intentional, then more rationale is needed. Also, what was the room temperature during the storage? If the purpose is to find out the effect of storage at room temperature, the study should have included samples stored at -80oC for comparison.
The purpose of this study was not to evaluate the effect of a storing litter at room temperature. Our assumption was based on the report obtained from farmers during sampling where they store litter for at least two duration before selling to crop farmers. Therefore, we made a hypothesis based on the storage condition which might help reduce the number of pathogens in
Reviewer 3 Report
Summary:
The present study highlights the selective pressure and the emergence of MDR through the use of antimicrobials in poultry farms. This study aimed to investigate the resistance profile of Escherichia coli isolated from poultry litter. The results indicate a resistance rate of 58.4% in E. coli isolates and a high prevalence of multiple resistance to antimicrobial agents used by human medicine. This study revealed that poultry bedding is a potential source of the spread of resistant germs from farm animals to the environment and to humans. The conclusions report that the improvement of biosecurity measures and the monitoring of bacterial resistance in farms is necessary to avoid the entry and spread of pathogens and the development of bacterial resistance, dangerous for human health. The following article was written in a fluid and correct form. It deals with an important issue in the field of environmental hygiene and safety. I recommend its publication.
Minor issues:
1) Table 2 and 3: please check the space after the comma, and check throughout the text, and correct the form 0-5- with 0-5 present in Duration in poultry farming (year) in table 2. Moreover, in Respect of vaccinal protocols, % in brackets has been added, but not to the other scores present, there is a specific reason or they can be deleted.
2) Improve the last part of table 3, Most common AMU used, it is not clear whether they are percentages of resistance, incidence or other; difficult to interpret by the reader.
3) In 2.2. Prevalence of antimicrobial….: « …of isolation of 44.1 % [95% CI : 37.7-50.5]. After a susceptibility testing to various antimicrobials, the E. coli strains isolated showed a global prevalence resistance of 58.4 % [95% CI : 48.8-68.0] … », please check the space after the percent.
4) Table 4: you can add the P-value in all result.
5) In figure 3: I don’t see the grey bar corresponding to IA results. The same problem in Figure 4, for Reserve result in red color.
6) Correct the font in this sentence «…farming, frequency of digestive tract infections, experience in poultry farming and respect of vaccine protocol were associated (p < 0.25) with…»; were, has a different size
7) In the Authors’ contributions : Conceptualization: M.M.M.M, J.A.N, A.W and F.M; Investigation and laboratory technical work: H.K.D, P.T, F.L.K. and A.W.; Data analysis: F.M, J.B.D, M.P.N and M.M.M.M., correct by adding dots to the letters.
Author Response
Minor issues:
1) Table 2 and 3: please check the space after the comma, and check throughout the text, and correct the form 0-5- with 0-5 present in Duration in poultry farming (year) in table 2. Moreover, in Respect of vaccinal protocols, % in brackets has been added, but not to the other scores present, there is a specific reason or they can be deleted.
Presentation of the number in parentheses was an error. This was modified and the % sign was deleted within the table 2
2) Improve the last part of table 3, Most common AMU used, it is not clear whether they are percentages of resistance, incidence or other; difficult to interpret by the reader.
The last part of table 3 reported the proportion of the antimicrobial use by farmers investigated. During the investigation, a specific question was addressed to identify the most antimicrobial agent used by the farmer. Each farmer has an option to make a multiple choice out of the panel of antimicrobial regularly use in Cameroon. The frequency of use was calculated as the ratio of the number of times a substance was selected by the total number of times all the substance was reported.
3) In 2.2. Prevalence of antimicrobial….: « …of isolation of 44.1 % [95% CI : 37.7-50.5]. After a susceptibility testing to various antimicrobials, the E. coli strains isolated showed a global prevalence resistance of 58.4 % [95% CI : 48.8-68.0] … », please check the space after the percent.
This was checked and modified accordingly
4) Table 4: you can add the P-value in all result.
5) In figure 3: I don’t see the grey bar corresponding to IA results. The same problem in Figure 4, for Reserve result in red color.
None of the molecule included in the IA or Reserve group was tested. Therefore, the corresponding bar was simply deleted.
6) Correct the font in this sentence «…farming, frequency of digestive tract infections, experience in poultry farming and respect of vaccine protocol were associated (p < 0.25) with…»; were, has a different size
This was directly modified within the text
7) In the Authors’ contributions : Conceptualization: M.M.M.M, J.A.N, A.W and F.M; Investigation and laboratory technical work: H.K.D, P.T, F.L.K. and A.W.; Data analysis: F.M, J.B.D., M.P.N and M.M.M.M., correct by adding dots to the letters.
This was modified accordingly in the manuscript